# Diagnostic Value of T2 Mapping in Sacroiliitis Associated with Spondyloarthropathy

**DOI:** 10.3390/diagnostics15131634

**Published:** 2025-06-26

**Authors:** Mustafa Koyun, Kemal Niyazi Arda

**Affiliations:** Department of Radiology, Gülhane Training and Research Hospital, University of Health Sciences, Ankara 06010, Türkiye; kemalarda@yahoo.com

**Keywords:** T2 mapping, T2 relaxation time, sacroiliac joint, sacroiliitis, osteitis

## Abstract

**Background/Objectives:** T2 mapping is a quantitative magnetic resonance imaging (MRI) technique that provides information about tissue water content and molecular mobility. This study aimed to evaluate the diagnostic utility of T2 mapping in assessing sacroiliitis associated with spondyloarthropathy (SpA). **Methods:** A prospective study examined a total of 56 participants, comprising 31 SpA patients (*n* = 31) and 25 healthy controls (*n* = 25), who underwent sacroiliac joint MRI between August 2018 and August 2020. T2 mapping images were generated using multi-echo turbo spin echo (TSE) sequence, and quantitative T2 relaxation times were measured from bone and cartilage regions. Statistical analysis employed appropriate parametric and non-parametric tests with significance set at *p* < 0.05. **Results:** The mean T2 relaxation time measured from the areas with osteitis of SpA patients (100.23 ± 7.41 ms; 95% CI: 97.51–102.95) was significantly higher than that of the control group in normal bone (69.44 ± 4.37 ms; 95% CI: 67.64–71.24), and this difference was found to be statistically significant (*p* < 0.001). No significant difference was observed between cartilage T2 relaxation times in SpA patients and controls (*p* > 0.05). **Conclusions:** T2 mapping serves as a valuable quantitative imaging biomarker for diagnosing sacroiliitis associated with SpA, particularly by detecting bone marrow edema. The technique shows promise for objective disease assessment, though larger studies are needed to establish standardized reference values for T2 relaxation times in osteitis to enhance diagnostic accuracy and facilitate treatment monitoring.

## 1. Introduction

The sacroiliac joint is the widest axial joint in the body, located between the sacrum and ilium bones, with an average surface area of 17.5 cm^2^, and it is defined as a diarthrodial synovial joint [1]. The sacroiliac joint consists of two parts with different joint characteristics. The first part is located along the anterior and inferior aspects of the joint, covered with cartilage, and has a synovial joint characteristic. The second part is located on the posterior and superior aspect, joined with the interosseous ligaments, and has a syndesmotic joint characteristic [2]. The sacroiliac joint is a complex structure that can develop a variety of degenerative changes during one’s life, including subchondral sclerosis, vacuum phenomenon, subchondral cyst, joint space narrowing, and ankylosis [3,4]. These age-related changes, combined with inflammatory conditions, can significantly affect joint function and cause chronic pain. Among the inflammatory conditions that commonly affect the sacroiliac joint, spondyloarthropathies (SpAs) represent the most clinically significant group.

SpAs represent a group of chronic rheumatic disorders with similar epidemiological, pathological, genetic, clinical, and radiological features [5]. This group consists of ankylosing spondylitis, arthritis associated with inflammatory bowel disease (IBD), reactive arthritis (previously Reiter syndrome), undifferentiated SpA, and psoriatic arthritis [6]. In 2009, the Assessment of Spondyloarthritis İnternational Society (ASAS) guidelines were published to provide standardization in the clinical and diagnostic approach to SpAs [7]. Magnetic resonance imaging (MRI) is a very crucial component of the ASAS criteria for evaluating sacroiliitis. MRI can evaluate not only the active findings of sacroiliitis such as osteitis, enthesitis, capsulitis, and synovitis, but also chronic findings such as subchondral sclerosis, periarticular fat deposition, periarticular erosions, and ankylosis [8]. The main sequences used in sacroiliac joint MRI are coronal and axial oblique fat-suppressed fast spin echo T2, short tau inversion recovery (STIR), spectral attenuated inversion recovery (SPAIR) sequences, and axial and coronal oblique plane fast spin echo T1-weighted sequences [8].

The earliest finding in sacroiliitis is osteitis [8]. Osteitis is the only criterion accepted by ASAS for the diagnosis of sacroiliitis, and the presence of other active inflammation findings (enthesitis, capsulitis, and synovitis) alone or together is not significant for sacroiliitis if there is no osteitis [8,9]. Osteitis is seen in almost 90% of SpA patients, but it can also be seen in other inflammatory (such as rheumatoid arthritis, Behçet’s disease, etc.) and infectious pathologies affecting the sacroiliac joint [10]. According to the ASAS criteria, in order to diagnose sacroiliitis, osteitis should be present in at least two different areas in a single slice or in at least two consecutive slices in the same area [7].

In recent years, various advanced imaging modalities have been introduced to quantitatively assess sacroiliitis in spondyloarthritis (SpA), including dynamic contrast-enhanced MRI (DCE-MRI), T2 mapping, diffusion-weighted imaging (DWI), and dual-energy computed tomography (DECT) with virtual non-calcium imaging capabilities [11]. Among these, DCE-MRI allows for the evaluation of tissue vascularization and the capillary network by measuring gadolinium concentration within tissues and blood vessels, thereby providing insights into inflammatory responses [12,13]. DWI, on the other hand, provides quantitative assessment of water molecule mobility within bone marrow, allowing for the detection of inflammatory cell infiltration and increased tissue water content in sacroiliac joints [14]. Meanwhile, DECT applies a three-material differentiation approach—distinguishing bone mineral, red marrow, and yellow marrow—to eliminate calcium content, thereby improving the visualization of anatomical details concealed by calcification and enabling the generation of virtual images of bone marrow without calcium [15]. Finally, among these quantitative techniques, T2 mapping is a promising MRI method that calculates the T2 relaxation times of various tissues and displays them on a parametric color map, offering objective tissue characterization beyond conventional morphological assessment.

T2 mapping images are generated by acquiring multiple echo time images from the same anatomical slice, then pixel-wise matching these images to calculate the T2 relaxation times for each corresponding location, ultimately creating a quantitative parametric map [16]. The T2 relaxation time is calculated using a weighted least squares fit with the natural logarithm of the region of interests (ROIs) mean signal [17]. Using the multi-echo spin echo (MESE), T2 prepared steady-state free precession (T2p-SSFP), and T2 gradient spin echo (T2-GraSE) sequences, a T2 mapping image can be created [18,19,20]. Independent of the imaging technique, a series of images are obtained from the same slice using multiple echo times. After processing these images with specialized software, a signal decay curve is generated for each pixel that eventually reflects the T2 relaxation times [21].

T2 relaxation times vary depending on the amount of water protons in the tissue and the anisotropy of the tissue matrix [22,23,24]. It is stated that the increase in the T2 relaxation time of a tissue is primarily related to an increased water content in that tissue, which reflects edema [18]. Given this sensitivity to tissue water content and structural changes, T2 mapping has found applications in various clinical scenarios. Research has shown that T2 mapping can be utilized to assess radiation-induced damage to the parotid gland [25], supraspinatus tendon pathologies [26], myocardial edema [27], and liver ischemia and reperfusion injury [28]. In addition to these various clinical applications, T2 mapping has gained significant attention in the field of joint and cartilage evaluation, being successfully applied to numerous joint cartilages, including those of the knee, interphalangeal, and glenohumeral joints [29,30,31]. T2 relaxation times are increased in damaged cartilage due to structural damage in the collagen tissue, decreased glycosaminoglycan content, increased water content, and increased water molecule motility [32,33,34]. It has been suggested that T2 mapping can be used as a non-invasive imaging method capable of detecting early-stage cartilage damage and monitoring treatment response in patients undergoing cartilage repair [35]. While T2 mapping has been extensively validated in other joints, research on the sacroiliac joint remains limited due to its complex anatomical structure, with only a few studies investigating this application [11,17,18,36,37,38]. Among these studies, some have focused exclusively on bone T2 relaxation times [11,36], whereas others have examined cartilage T2 relaxation times [17,18,37,38] separately. Our study, however, aimed to evaluate the diagnostic potential of T2 mapping in detecting sacroiliitis associated with SpA by providing a comprehensive analysis through the assessment of T2 relaxation times from both bone and cartilage regions.

## 2. Materials and Methods

### 2.1. Study Population

This study was approved by the Health Sciences University Non-Interventional Clinical Research Ethics Committee (Decision No: 18/146, Date: 22 May 2018). A prospective evaluation was performed on 56 adult participants referred from various clinics to our radiology unit between August 2018 and August 2020. All participants were informed about the study procedures, and written informed consent was obtained. Prior to MRI, demographic data (e.g., age, sex), clinical data, laboratory findings, and any relevant pathological diagnoses were recorded and securely stored.

The SpA group included patients whose symptoms (inflammatory back pain, arthritis, enthesitis) and laboratory findings (C-reactive protein [CRP], human leukocyte antigen-B27 [HLA-B27]) satisfied the ASAS criteria, along with evidence of osteitis on conventional MRI. The healthy control group comprised individuals over 18 years of age who did not meet any ASAS criteria, had no evidence of infective-rheumatologic sacroiliitis, had no history of sacroiliac joint trauma, presented with normal laboratory results, and exhibited no radiological signs of sacroiliitis on MRI.

Exclusion criteria consisted of claustrophobia, the presence of medical devices incompatible with MRI, and images deemed uninterpretable due to artifacts.

### 2.2. MRI Protocol

All MRI examinations were conducted on a 3 Tesla MRI scanner (Achieva; Philips Medical Systems, Best, Netherlands) equipped with a 32-channel torso coil. In addition to the routine MRI sequences employed for sacroiliac joint imaging (Table 1), an axial oblique multi-echo turbo spin echo (TSE) sequence was used to generate T2 mapping images (Table 2).

### 2.3. Image Processing and T2 Relaxation Time Measurement

Due to the lack of dedicated T2 mapping software on our MRI workstation, the open-source software MRmap version 1.4 was used to generate T2 mapping images [40,41]. T2 mapping images were generated from source images acquired with a multi-echo TSE pulse sequence, with specific TE values used for each image (Figure 1). T2 calculations were performed using a 2-parameter Levenberg–Marquardt curve fitting procedure [40].

T2 mapping images were created from selected slices from the upper, middle, and lower portions of the joint in both SpA and control groups. In this context, a total of 168 T2 mapping images (56 cases × 3 slices) were included in the evaluation. Since the software allowed pixel-level sampling, multiple measurements (five measurements per region) were performed from the relevant regions to enhance measurement accuracy, and their median values were calculated.

In the SpA group, T2 relaxation times were measured from areas with osteitis and their adjacent cartilage (Figure 2). Measurements from areas with osteitis were performed from regions where edema was most prominent. In the control group, T2 relaxation times were measured from normal bone (sacral and iliac parts) and cartilage from both anterior and posterior portions of the joint in each slice (Figure 3). Since the syndesmotic regions of the sacroiliac joint do not contain cartilage, no measurements were performed from these areas. To avoid affecting the measurements, no measurements were taken from severely damaged cartilage that could not be evaluated near areas with osteitis, or from areas suspected of vacuum phenomena/joint fluid. In total, T2 relaxation times were measured from 268 areas with osteitis in the SpA group and from 600 different normal bone regions (25 cases × 2 sacroiliac joints × 12 bone regions) in the control group. T2 relaxation times were measured from 70 cartilage regions adjacent to areas with osteitis in the SpA group and from 200 cartilage regions in the control group.

### 2.4. Statistical Analysis

All statistical analyses were performed using IBM SPSS 25 software (IBM Corp., Armonk, NY, USA). Normality was tested using the Kolmogorov–Smirnov test. For groups with normally distributed data, the independent samples *t*-test was applied, whereas the Mann–Whitney U test was used for groups not meeting the normality assumption. A Chi-square test was conducted to compare gender distribution. All continuous data were reported as mean ± standard deviation (Mean ± SD) regardless of distribution, whereas categorical data were described using frequencies (n) and percentages (%). *p*-value < 0.05 was considered statistically significant.

## 3. Results

A total of 56 participants were included in this study, comprising 31 SpA patients (15 female and 16 male) and 25 healthy controls (11 female and 14 male). The mean age of the control group was 37.2 ± 9.6 years, while the mean age of the SpA group was 35.4 ± 13.0 years (Table 3). No statistically significant difference was observed between the SpA and control groups with respect to sex or age (*p* > 0.05).

The frequencies of ASAS criteria observed among SpA patients are presented in Table 4. All SpA patients demonstrated active sacroiliitis on conventional MRI. Moreover, elevated CRP levels (77.4%) and inflammatory back pain (61.2%) were more prevalent than the other criteria in these patients. Crohn’s disease or ulcerative colitis was not identified in any patient.

Table 5 shows the mean T2 relaxation times, measured through T2 mapping, of bone and cartilage tissues in both the SpA and control groups. The mean T2 relaxation time for areas with osteitis in SpA patients (100.23 ± 7.41 ms; 95% CI: 97.51–102.95) was significantly higher than that of the control group (69.44 ± 4.37 ms; 95% CI: 67.64–71.24) (*p* < 0.001) (Table 5). No significant difference was detected between the mean T2 relaxation time of cartilage adjacent to areas with osteitis in SpA patients (44.0 ± 3.19 ms; 95% CI: 42.83–45.17) and that of cartilage in the control group (43.2 ± 3.41 ms; 95% CI: 41.79–44.61) (*p* = 0.249) (Table 5).

## 4. Discussion

In this study, we investigated the diagnostic potential of T2 mapping imaging in patients with SpA sacroiliitis. In our literature review, we have found that there are limited number of T2 mapping studies conducted for the sacroiliac joint [11,17,18,36,37,38,42].

Findings of this study were comparable to previous reports investigating the effect of SpA on T2 relaxation time. Specifically, we determined that the mean T2 relaxation times of the areas with osteitis (100.23 ± 7.41 ms) were higher compared to the healthy control group (69.44 ± 4.37 ms), which aligns with the elevated T2 values in areas with osteitis reported by Wang et al. and Zhang et al. Wang et al. found T2 relaxation times of 123.77 ± 15.85 ms in osteitis areas compared to 89.34 ± 10.65 ms in the healthy control group [36], while Zhang et al. reported values of 104.7 ms in osteitis areas versus 91.8 ± 14.3 ms in normal bone tissue [11]. The variation in these T2 values across the literature may reflect differences in inflammation severity among patient populations in different studies, as higher T2 values indicate stronger active inflammation [36,43]. The elevated T2 values in osteitis areas likely result from increased free water content secondary to inflammatory exudation and cellular infiltration, making T2 relaxation time a valuable biomarker for assessing disease activity [44].

Previous studies by Lefebvre et al. and Albano et al. have established the feasibility and reliability of T2 mapping imaging methodology in healthy populations [17,18]. In the study by Lefebvre et al., the mean T2 values measured by two independent observers were 41.9 ms and 40.8 ms, with an overall mean of 41.6 ms. Similarly, Albano et al. reported T2 values of 42 ms and 40.7 ms from two observers, with a combined mean of 41.1 ms [17,18]. The mean cartilage T2 relaxation time observed in our healthy control group (43.2 ± 3.41 ms) was in accordance with previously established values in the literature. Albano et al. reported in another study that SpA patients exhibited higher mean T2 relaxation times in sacroiliac cartilage (58.5 ± 4.4 ms) relative to healthy individuals (44.1 ± 6.6 ms) [42]. Similarly, Kasar et al. demonstrated that cartilage T2 relaxation times in SpA patients (50.48 ± 5.32 ms) were significantly higher compared to the control group (46.33 ± 3.30 ms), with a statistically significant difference observed between the groups [37]. Conversely, Francavilla et al. reported no statistically significant difference in cartilage T2 relaxation times between sacroiliitis patients (43.04 ms; IQR: 41.25–49.76 ms) and controls (40.0 ms; IQR: 38.9–48.6 ms) [38]. Similarly to Francavilla et al., our study revealed no statistically significant difference in mean cartilage T2 relaxation times between the healthy control and SpA groups. This result can be explained by our methodology of excluding severely damaged cartilage tissue in the vicinity of areas with osteitis that precluded accurate assessment.

Radiography, CT, and MRI are commonly used in the assessment of the sacroiliac joint in current clinical practice. With radiography and CT imaging techniques, structural changes that appear later, such as subchondral sclerosis, periarticular erosions, joint space narrowing and ankylosis can be reliably demonstrated in the sacroiliac joint [2,45]. Although radiography and CT imaging methods can reliably reveal late-stage structural changes, they are not successful in detecting active lesions of sacroiliitis in the early stages. Therefore, delays can occur in the treatment of sacroiliitis. On the other hand, MRI cannot only reveal late-stage structural changes but also successfully demonstrate active lesions in the early stages of sacroiliitis [8]. Today, the success of MRI in diagnosing sacroiliitis is indisputable. According to the ASAS criteria, presence of bone marrow edema, or osteitis in other words, is diagnostic parameter for sacroiliitis [10]. In the demonstration of the osteitis, fat-suppressed T2-weighted images such as STIR, SPAIR, and fat-suppressed contrast-enhanced T1-weighted images are used. However, both radiography/CT and conventional MRI provide us with the opportunity for qualitative evaluation of the sacroiliac joint. T2 mapping, on the other hand, provides quantitative data in osteitis assessment. The clinical potential of T2 mapping extends beyond simple detection of osteitis. This quantitative approach offers several advantages over conventional imaging methods: First, T2 mapping can provide objective, numerical values that may enhance diagnostic accuracy and reduce inter-observer variability commonly encountered in qualitative MRI assessment. Second, the quantitative nature of T2 mapping makes it particularly valuable for monitoring disease progression and treatment response, as subtle changes in tissue composition can be detected through T2 relaxation time measurements even before they become visually apparent on conventional sequences. Third, T2 mapping may serve as a biomarker for early inflammatory changes, potentially identifying subclinical osteitis before it reaches the threshold for visual detection on standard MRI sequences. Fourth, T2 mapping is a non-invasive imaging technique, which does not require any contrast agent injection [46,47]. Additionally, the absence of radiation exposure in T2 mapping imaging can be considered as another advantage. In clinical practice, T2 mapping could be integrated into routine MRI protocols to provide quantitative assessment, potentially improving diagnostic confidence, especially in borderline cases where conventional MRI findings may be ambiguous.

Despite its usefulness in quantification, T2 mapping also has some disadvantageous aspects. The first of these is that obtaining the multi-echo images required to generate T2 mapping images and processing these images is quite time-consuming. In our study, it took 6 min and 53 s to obtain multi-echo images for each patient. Lefebvre and colleagues collected data in 12 min and 49 s for each patient, while Albano and colleagues performed the task in 5 min and 42 s [17,18]. We predict that with the new technological developments in the future, T2 mapping imaging can be performed in a shorter time. Also, the multi-echo sequence required for generating T2 mapping images is very sensitive to motion artifacts, which can negatively affect image processing [48]. In addition, interpretation of T2 mapping images is operator dependent, which can also affect, the measured T2 relaxation times [49].

There are some limitations with this study. Firstly, the morphologic characteristics of the sacroiliac joint posed challenges, as the restricted area made it difficult to examine the cartilage components thoroughly. Secondly, although we attempted to avoid them during measurements, the vacuum phenomenon and joint fluid, which are frequently observed in the sacroiliac joint, may have affected our T2 measurements. Thirdly, despite performing multiple measurements and taking their average, our findings may have been influenced by our pixel-level sampling approach. Fourthly, potential calculation errors due to the partial volume effect of surrounding tissues may have affected our results. Fifthly, the absence of inter-observer repeatability assessment represents a limitation of our methodology. Finally, our study was carried out with a small sample size, and a reference T2 relaxation time could not be determined for osteitis. Working with larger patient groups is necessary to establish a reference T2 relaxation time for osteitis.

## 5. Conclusions

In conclusion, T2 mapping imaging was evaluated as a successful imaging technique for distinguishing areas with osteitis from normal bone tissue. The findings of this study demonstrated that the T2 mapping can be used as a tool for the diagnosis of sacroiliitis associated with SpA. In the future, working with larger patient groups and determining a reference T2 relaxation value for osteitis through T2 mapping imaging may be beneficial in both diagnosing SpA sacroiliitis and evaluating the effectiveness of applied treatments.

## Figures and Tables

**Figure 1 diagnostics-15-01634-f001:**
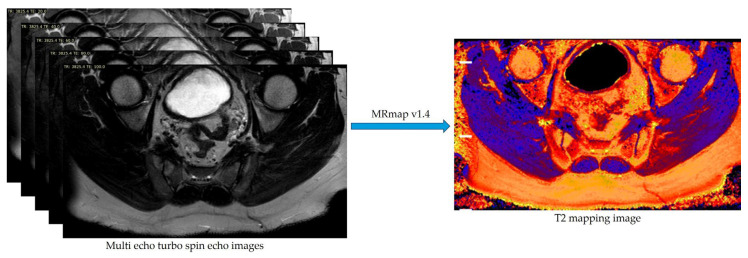
Conversion of multi-echo turbo spin echo images to a T2 mapping image using MRmap v1.4 software.

**Figure 2 diagnostics-15-01634-f002:**
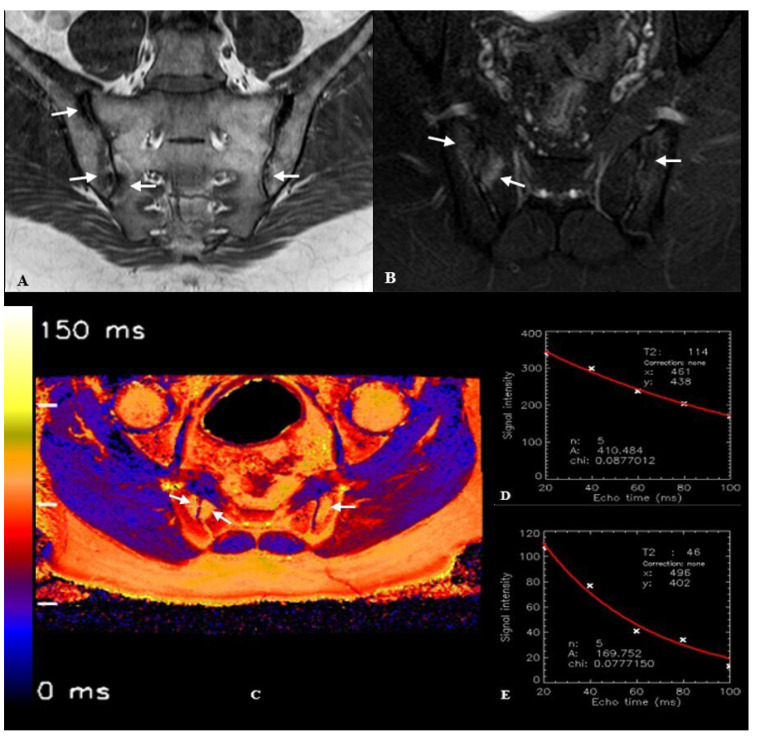
Images of a 31-year-old male patient with spondyloarthritis (SpA)-associated sacroiliitis (figure from [39]). (**A**). Coronal oblique turbo spin echo (TSE) T1-weighted image of the sacroiliac joints demonstrates hypointense areas representing bone marrow edema/osteitis in bilateral sacroiliac joints (white arrows). (**B**). Axial oblique TSE spectral attenuated inversion recovery (SPAIR) image obtained through the lower sacroiliac joint level reveals hyperintense signal abnormalities consistent with osteitis, particularly prominent in the right sacroiliac joint (white arrows). (**C**). Axial oblique T2 mapping image acquired at the corresponding lower sacroiliac joint level, with osteitis indicated by white arrows. (**D**). T2 relaxation time curve derived from the sacral component of the right sacroiliac joint, demonstrating an elevated T2 relaxation time of 114 ms. (**E**). T2 relaxation time curve obtained from the anterior cartilaginous region of the right sacroiliac joint, showing a T2 relaxation time of 46 ms.

**Figure 3 diagnostics-15-01634-f003:**
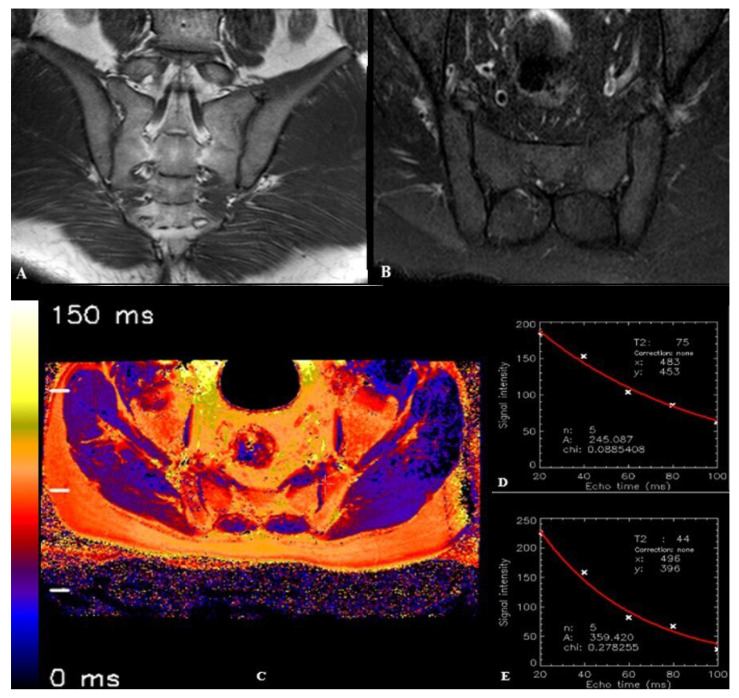
Images of a 40-year-old male patient with normal sacroiliac joint anatomy (figure from [39]). (**A**). Coronal oblique turbo spin echo (TSE) T1-weighted image demonstrates normal morphology of bilateral sacroiliac joints. (**B**). Axial oblique TSE spectral attenuated inversion recovery (SPAIR) image through the lower sacroiliac joint level shows normal signal characteristics in both sacroiliac joints without evidence of bone marrow edema or structural abnormalities. (**C**). Axial oblique T2 mapping image obtained at the corresponding lower sacroiliac joint level. (**D**). T2 relaxation time curve derived from the sacral component of the left sacroiliac joint, demonstrating a T2 relaxation time of 75 ms. (**E**). T2 relaxation time curve obtained from the anterior cartilaginous region of the left sacroiliac joint, showing a T2 relaxation time of 44 ms.

**Table 1 diagnostics-15-01634-t001:** Sequences and their parameters used in routine sacroiliac joint MRI (data from [39]).

Parameters	T1-Weighted TSE	SPAIR TSE
Plane	Coronal oblique	Axial/Coronal oblique
TE (ms)	8	80
TR (ms)	593	3431
FOV (mm)	200	200
Flip angle (°)	90	90
No. of signal averages	2	2
Slice thickness (mm)	4	4
Echo train length	6	19

FOV: field of view; TE: time to echo; TR: repetition time; TSE: turbo spin echo; SPAIR: spectral attenuated inversion recovery.

**Table 2 diagnostics-15-01634-t002:** Parameters of the multi-echo TSE sequence used to generate T2 mapping images (data from [39]).

Parameters	Multi-Echo TSE
Plane	Axial oblique
TE (ms)	20, 40, 60, 80, 100
TR (ms)	3825
No. of echoes	5
Echo spacing (ms)	20
FOV (mm)	200
Flip angle (°)	90
No. of signal averages	1
Slice thickness (mm)	4
Bandwidth (pixels)	219
Acquisition matrix	268 × 265
Acquisition time	6.53 min
Acquired pixel resolution (mm)	0.75 × 0.75

FOV: field of view; TE: time to echo; TR: repetition time; TSE: turbo spin echo.

**Table 3 diagnostics-15-01634-t003:** Demographic characteristics (gender and age) of the control and SpA groups (data from [39]).

Gender	SpA Group (*n* = 31)	Control Group (*n* = 25)	Total (*n* = 56)	* p * Value
Female, *n* (%)	15 (48.3)	11 (44)	26 (100)	>0.05
Male, *n* (%)	16 (51.7)	14 (56)	30 (100)	>0.05
Age, Mean ± SD	35.4 ± 13.0	37.2 ± 9.6	36.2 ± 12.0	>0.05

SD: standart deviation; SpA: spondyloarthropathy.

**Table 4 diagnostics-15-01634-t004:** The frequencies of ASAS criteria observed in SpA patients (data from [39]).

ASAS Criteria		*n* (%)
Inflammatory Back Pain		19 (61.2)
Arthritis		12 (38.7)
Enthesitis		4 (12.9)
Uveitis		2 (6.4)
Dactylitis		3 (9.6)
Psoriasis		2 (6.4)
Crohn’s/Ulcerative Colitis		0 (0)
Family history for SpA		6 (19.3)
HLA-B27	Negative	16 (51.6)
	Positive	7 (22.6)
	Unknown	8 (25.8)
Good Response to NSAIDs	No	5 (16.1)
	Yes	12 (38.7)
	Unknown	14 (45.2)
CRP Level	<5 mg/L	7 (22.6)
	>5 mg/L	24 (77.4)
Active Sacroiliitis on MRI		31 (100)

ASAS: Assessment of Spondyloarthritis International Society; CRP: C-reactive protein; HLA: human leukocyte antigen; MRI: magnetic resonance imaging; NSAIDs: nonsteroidal antiinflammatory drugs; SpA: spondyloarthropathy.

**Table 5 diagnostics-15-01634-t005:** Comparison of the mean T2 relaxation times of bone and cartilage tissues obtained from the SpA and control groups (data from [39]).

	SpA Group (*n* = 31)	Control Group (*n* = 25)	*p* Value
T2 relaxation time—bone (ms) *, (Mean ± SD)	100.23 ± 7.41	69.44 ± 4.37	<0.001
T2 relaxation time—cartilage (ms) **, (Mean ± SD)	44.0 ± 3.19	43.2 ± 3.41	0.249

* Independent samples *t*-test; ** Mann–Whitney U Test; SD: standard deviation; SpA: spondyloarthropathy.

## Data Availability

The data presented in this study are available upon request from the corresponding author. The data are not publicly available due to privacy. The figures (Figure 2 and Figure 3) and all data tables are taken from the first author’s Medical Specialty thesis [39], which is available through the Council of Higher Education National Thesis Center. The use of thesis data complies with the provisions of the Higher Education Law No. 2547 for scientific research purposes.

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
