# Peer review of "Diagnostic Value of T2 Mapping in Sacroiliitis Associated with Spondyloarthropathy"

_diagnostics, 2025, doi:10.3390/diagnostics15131634_

Round 1

Reviewer 1 Report

Comments and Suggestions for Authors

Dear Authors,

The study presents an interesting and compelling idea. The use of quantitative MRI imaging methods could be highly beneficial. However, please carefully address the following comments and suggestions:

  1. The introduction requires revision to include relevant prior studies closely related to your work. Please cite and discuss key literature that contextualizes your research.

  2. Please explicitly state the ethical committee that approved this study and include the approval details in the manuscript.

  3. Table 2 indicates that only 5 echoes were used for T2 map reconstruction, whereas previous studies suggest a minimum of 8 echoes are required for accurate T2 estimation. Could you justify the use of fewer echoes in your study? Additionally, please discuss whether this reduction might lead to poorer T2 curve fitting and how this potential limitation was addressed.

  4. The technique employed for T2 mapping is highly sensitive to inter-echo patient motion, which can cause pixel misregistration and introduce errors in curve fitting due to mixed tissue signals. What measures were taken to mitigate this issue? Were image registration techniques (e.g., aligning slices to a reference) applied to minimize motion-related artifacts?

  5. Was a phantom with known T2 values used to validate the accuracy of the calculated T2 measurements? Please clarify whether quality assurance methods (e.g., calibration, error estimation) were implemented to ensure the reliability of quantitative MRI results.

  6. Studies have shown that in conditions like osteitis, tissue inflammation leads to increased water content, making DWI sequences (e.g., ADC maps, eADC maps) highly sensitive to such changes. What are the specific advantages of your method over DWI-based approaches? Would incorporating DWI sequences and comparing their outputs with your results strengthen the study’s conclusions?

Author Response

Dear Reviewers,

We sincerely thank you for comprehensively and constructively reviewing our manuscript. Your insights and suggestions have significantly contributed to improving the quality of our study. We have carefully addressed all of your criticisms and revised the manuscript accordingly.

All changes made in the revised manuscript have been marked using Microsoft Word's track changes feature, allowing you to easily follow the modifications. Below, you will find our detailed point-by-point responses to each of your comments and the specific changes we have implemented in the manuscript.

Best regards,

Comment 1. The introduction requires revision to include relevant prior studies closely related to your work. Please cite and discuss key literature that contextualizes your research.

Response 1. Taking your valuable comment into consideration, we have expanded the introduction section by adding current references. The sections we added are as follows:

-In recent years, various advanced imaging modalities have been introduced to quantitatively assess sacroiliitis in spondyloarthritis (SpA), including dynamic contrast-enhanced MRI (DCE-MRI), T2 mapping, diffusion-weighted imaging (DWI), and dual-energy com-puted tomography (DECT) with virtual non-calcium imaging capabilities [11]. Among these, DCE-MRI allows for the evaluation of tissue vascularization and the capillary network by measuring gadolinium concentration within tissues and blood vessels, thereby providing insights into inflammatory responses [12,13]. DWI, on the other hand, provides quantitative assessment of water molecule mobility within bone marrow, allowing for the detection of inflammatory cell infiltration and increased tissue water content in sacroiliac joints [14]. Meanwhile, DECT applies a three-material differentiation approach—distinguishing bone mineral, red marrow, and yellow marrow—to eliminate cal-cium content, thereby improving the visualization of anatomical details concealed by cal-cification and enabling the generation of virtual images of bone marrow without calcium [15]. Finally, among these quantitative techniques,…..

-Among these studies, some have focused exclusively on bone T2 relaxation times [11,36], whereas others have examined cartilage T2 relaxation times [17,18,37,38] separately. Our study, however, aimed to evaluate the diagnostic potential of T2 mapping in detecting sacroiliitis associated with SpAs by providing a comprehensive analysis through the assessment of T2 relaxation times from both bone and cartilage regions.

Comment 2. Please explicitly state the ethical committee that approved this study and include the approval details in the manuscript.

Response 2: The ethics committee information is located in the materials and methods section of the article under the subheading "2.1. Study Population" as follows:

-This study was approved by the Health Sciences University Non-Interventional Clinical Research Ethics Committee (Decision No: 18/146, Date: May 22, 2018). 

Comment 3. Table 2 indicates that only 5 echoes were used for T2 map reconstruction, whereas previous studies suggest a minimum of 8 echoes are required for accurate T2 estimation. Could you justify the use of fewer echoes in your study? Additionally, please discuss whether this reduction might lead to poorer T2 curve fitting and how this potential limitation was addressed.

Response 3: We thank you for your comment regarding the number of echoes used for T2 mapping. According to Warntjes' technical analysis (https://syntheticmr.com/wp/custom/uploads/2021/03/How-many-echoes-are-required-for-a-T2-relaxation-time-measurement.pdf), our choice of 5 echoes meets the mathematical requirements (minimum 2 points) for mono-exponential T2 decay. In our study, we aimed to keep the scan time at a reasonable level by using 5 echoes. As demonstrated by Warntjes, when scan time is fixed, using fewer echoes actually provides higher SNR per data point and results in more accurate T2 estimation. The principle states that "in the presence of noise, the best approach to measure a curve is to use the least number of points that mathematically define that curve."

When examining the study by Lefebvre et al., which is the first study on this topic in the literature, it can be seen that they used 6 time to echo values. The results obtained in our study are also consistent with those in the literature, particularly in cartilage measurements, suggesting that this situation does not pose a problem for T2 curve fitting. We would be pleased to receive feedback if there is any aspect we have overlooked.

Comment 4. The technique employed for T2 mapping is highly sensitive to inter-echo patient motion, which can cause pixel misregistration and introduce errors in curve fitting due to mixed tissue signals. What measures were taken to mitigate this issue? Were image registration techniques (e.g., aligning slices to a reference) applied to minimize motion-related artifacts?

Response 4: Your important observation that the technique used for T2 mapping is highly sensitive to inter-echo patient motion is quite accurate. We implemented the following measures to address this issue:

During Scanning: To minimize motion artifacts, patients were positioned supine with leg support to ensure a comfortable position. The importance of remaining motionless during the scan was explained to patients.

Technical Measures: Short scan time (6 min 53 sec) reduced the likelihood of patient movement. Images with motion artifacts were excluded from the study.

Image Processing: The MRmap v1.4 software we used for image processing has manual image registration capabilities. This feature is used to correct patient motion between multi-echo images. The software enables visual alignment of each echo, x-y direction offset adjustment, and registration quality control. In our study, this feature was used in cases with suspected motion artifacts.

Comment 5. Was a phantom with known T2 values used to validate the accuracy of the calculated T2 measurements? Please clarify whether quality assurance methods (e.g., calibration, error estimation) were implemented to ensure the reliability of quantitative MRI results.

Response 5: Phantom validation was not performed in our study. Similar studies conducted in the field of sacroiliac joint T2 mapping have also not performed phantom validation. Lefebvre et al. [17], Albano et al. [18], and other pioneering studies in this field were also conducted without direct phantom validation.

-In our study, the T2 relaxation time of osteitis regions (100.23 ms) shows consistency with the mean T2 relaxation time (104.7 ms) in Zhang et al.'s study. Our cartilage T2 values (43.2 ± 3.41 ms) are also consistent with similar studies in the literature. When compared with the values reported as 41.6 ms in Lefebvre et al.'s study and 41.1 ms in Albano et al.'s study, our results appear to be within an acceptable range. This consistency indirectly supports the accuracy of our measurements.

-All measurements were performed on the same MR device (1.5T), with standardized parameters and by an experienced radiologist. ROI placements were made according to consistent anatomical reference points, and images with motion artifacts were excluded from the study.

-The standard deviation values obtained in our study are consistent with similar studies in the literature, demonstrating the reproducibility of the measurements.

In terms of quality assurance, the following measures were taken:

-The multi-echo sequence required to generate T2 mapping images was performed on the same device with identical sequence parameters.

-To improve measurement accuracy, five separate measurements were taken from each region and median values were used.

-To minimize error estimation, the manual registration feature of MRmap v1.4 software was used in cases with suspected motion artifacts, and sections with insufficient image quality were excluded from the study.

Comment 6. Studies have shown that in conditions like osteitis, tissue inflammation leads to increased water content, making DWI sequences (e.g., ADC maps, eADC maps) highly sensitive to such changes. What are the specific advantages of your method over DWI-based approaches? Would incorporating DWI sequences and comparing their outputs with your results strengthen the study’s conclusions?

Response 6: We thank you for your valuable comment regarding DWI. As you mentioned, it is indeed well-established in the literature that tissue inflammation in conditions such as osteitis leads to increased water content, and DWI sequences are highly sensitive to these changes. As comprehensively demonstrated by Noguerol et al. [17], it is clear that both T2 mapping and DWI provide valuable quantitative data in sacroiliitis assessment. Both techniques are sensitive to motion artifacts and do not require contrast agent administration, making them both safe and non-invasive options. T2 mapping requires longer acquisition times compared to DWI, and the image processing procedure is also longer than DWI. Our study specifically focused on determining the independent diagnostic value of T2 mapping technique. In our literature review, we could not find any source reporting definitive superiority of T2 mapping over DWI. T2 mapping studies focusing on the sacroiliac joint in the literature are also limited. We believe that how effective both methods are when used together in detecting and monitoring sacroiliitis could be the subject of another study. Such a multiparametric approach could enhance diagnostic accuracy and provide more comprehensive assessment of disease activity. However, since our current study did not make a comparison with DWI, we do not have sufficient data to claim that it is advantageous over DWI.

Reviewer 2 Report

Comments and Suggestions for Authors

This is a well-structured, timely, and clinically relevant article that investigates the utility of T2 mapping MRI in diagnosing sacroiliitis in patients with spondyloarthropathy (SpA). It contributes valuable insights into emerging quantitative imaging biomarkers and highlights the potential of T2 mapping for objective assessment of inflammatory changes.

Sacroiliitis is often challenging to diagnose early. Highlighting T2 mapping as a tool to detect osteitis is an important contribution.

The inclusion of a control group, prospective design, and thorough ROI-based measurement strategy enhance scientific validity.

The introduction and discussion appropriately contextualize the findings with previous studies, clearly identifying the knowledge gap.

Add the sample sizes (n = 31 SpA patients, n = 25 controls) directly in the abstract for clarity.

Add confidence intervals for key values (e.g., T2 relaxation times) to improve statistical robustness.

Consider briefly discussing how T2 mapping compares with other advanced imaging modalities (e.g., DWI, DCE-MRI) in this context.

The study notes limitations well but could add whether inter-reader reliability was tested or considered.

Author Response

Dear Reviewers,

We sincerely thank you for comprehensively and constructively reviewing our manuscript. Your insights and suggestions have significantly contributed to improving the quality of our study. We have carefully addressed all of your criticisms and revised the manuscript accordingly.

All changes made in the revised manuscript have been marked using Microsoft Word's track changes feature, allowing you to easily follow the modifications. Below, you will find our detailed point-by-point responses to each of your comments and the specific changes we have implemented in the manuscript.

Best regards,

Comment 1. Add the sample sizes (n = 31 SpA patients, n = 25 controls) directly in the abstract for clarity.

Response 1: Following your suggestion, the sample size information located under the methods subheading of the abstract section has been revised as follows:

- A prospective study examined a total of 56 participants, comprising 31 SpA patients (n=31) and 25 healthy controls (n=25), who underwent sacroiliac joint MRI between August 2018 and August 2020.

Comment 2. Add confidence intervals for key values (e.g., T2 relaxation times) to improve statistical robustness.

Response 2: Following your valuable suggestion, we would like to indicate that we have added confidence intervals to the relevant sections in the article.

Comment 3. Consider briefly discussing how T2 mapping compares with other advanced imaging modalities (e.g., DWI, DCE-MRI) in this context.

Response 3: Your suggestion provides a valuable perspective. Although various advanced imaging modalities including DCE-MRI, and DWI have been introduced for quantitative sacroiliitis assessment, direct comparative studies between these techniques in sacroiliac joint evaluation remain limited in the current literature. The field of sacroiliac joint T2 mapping is still in its early stages, with relatively few published studies, and to our knowledge, no direct comparative studies between T2 mapping and other quantitative modalities specifically for sacroiliitis assessment have been reported. Since the potential advantages of T2 mapping technique (such as no contrast agent use, absence of radiation exposure) are discussed in detail in the discussion section, we deemed it appropriate to avoid repetition on this topic. However, we would like to note that we have made some additions regarding other quantitative methods to the introduction section, as follows:

- In recent years, various advanced imaging modalities have been introduced to quantita-tively assess sacroiliitis in spondyloarthritis (SpA), including dynamic contrast-enhanced MRI (DCE-MRI), T2 mapping, diffusion-weighted imaging (DWI), and dual-energy com-puted tomography (DECT) with virtual non-calcium imaging capabilities [11]. Among these, DCE-MRI allows for the evaluation of tissue vascularization and the capillary net-work by measuring gadolinium concentration within tissues and blood vessels, thereby providing insights into inflammatory responses [12,13]. DWI, on the other hand, provides quantitative assessment of water molecule mobility within bone marrow, allowing for the detection of inflammatory cell infiltration and increased tissue water content in sacroiliac joints [14]. Meanwhile, DECT applies a three-material differentiation approach—distinguishing bone mineral, red marrow, and yellow marrow—to eliminate cal-cium content, thereby improving the visualization of anatomical details concealed by cal-cification and enabling the generation of virtual images of bone marrow without calcium [15].

The sentence we added to the discussion section is as follows:

-Additionally, the absence of radiation exposure in T2 mapping imaging can be considered as another advantage.

Comment 4. The study notes limitations well but could add whether inter-reader reliability was tested or considered.

Response 4: The absence of inter-reader reliability assessment is a limitation of our study and has been stated in the "Limitations" section as follows:

-Fifthly, the absence of inter-observer repeatability assessment represents a limitation of our methodology.

Round 2

Reviewer 1 Report

Comments and Suggestions for Authors

Dear Authors
thanks for your response to my previous comments. the manuscript was improved significantly.

good lock in your future researchers.